# A Hydroquinone-Based Derivative Elicits Apoptosis and Autophagy via Activating a ROS-Dependent Unfolded Protein Response in Human Glioblastoma

**DOI:** 10.3390/ijms20153836

**Published:** 2019-08-06

**Authors:** Silvia Zappavigna, Alessia Maria Cossu, Marianna Abate, Gabriella Misso, Angela Lombardi, Michele Caraglia, Rosanna Filosa

**Affiliations:** 1Department of Precision Medicine, University of Campania “Luigi Vanvitelli”, via L. De Crecchio 7, 80138 Naples, Italy; 2Biogem Scarl, Institute of Genetic Research, Laboratory of Precision and Molecular Oncology, Contrada Camporeale, 83031 Ariano Irpino (AV), Italy; 3Department of Experimental Medicine, University of Campania “Luigi Vanvitelli”, via L. De Crecchio 7, 80138 Naples, Italy; 4Consorzio Sannio Tech-AMP Biotec, Appia Str. 7, BN 82030 Apollosa, Italy; 5Institute of Food Sciences, National Research Council, Roma Str. 64, 83100 Avellino, Italy

**Keywords:** anticancer drug, apoptosis, autophagy, ER stress, natural compound, reactive oxygen species (ROS)

## Abstract

5-Lipoxygenase (5-LO) has been reported to be highly expressed in brain tumors and to promote glioma cell proliferation. Therefore, we investigated the anticancer activity of the novel 5-LO inhibitor derivative 3-tridecyl-4,5-dimethoxybenzene-1,2-diol hydroquinone (EA-100C red) on glioblastoma (GBM) cell growth. Cell viability was evaluated by MTT assay. The effects of the compound on apoptosis, oxidative stress and autophagy were assessed by flow cytometry (FACS). The mode of action was confirmed by Taqman apoptosis array, Real Time qPCR, confocal microscopy analysis and the western blotting technique. Our results showed that EA-100C Red had a higher anti-proliferative effect on LN229 as compared to U87MG cells. The compound induced a significant increase of apoptosis and autophagy and up-regulated pro-apoptotic genes (Bcl3, BNIP3L, and NFKBIA) in both GBM cell lines. In this light, we studied the effects of EA-100C red on the expression of CHOP and XBP1, that are implicated in ER-stress-mediated cell death. In summary, our findings revealed that EA-100C red induced ER stress-mediated apoptosis associated to autophagy in GBM cells through CHOP and Beclin1 up-regulation and activation of caspases 3, 9, JNK and NF-kappaB pathway. On these bases, EA-100C red could represent a promising compound for anti-cancer treatment.

## 1. Introduction

Glioblastoma (GBM) is one of the most frequent malignant cancers occurring in the human brain. The association of temozolomide (TMZ) with radiotherapy is the ideal first-line therapy for GBM [1]. Despite of this hard line treatment, and the latest improvements obtained with the combination of chemotherapy, surgery, and radiations, GBM regularly recurs and clinical outcome is poor [2,3]. Recently, fotemustine (FTM) has shown a significant action on recurrent GBM in phase II trials [4,5]. FTM is a highly lipophilic nitrosourea that may cross the blood–brain barrier (BBB) [6,7]. FTM also showed a significant antitumor activity on human GBM in preclinical models [8]. In spite of a wide range of recent specific treatments, GBM therapy remains a challenge. The BBB is the main obstacle for the advances in the delivery of new drugs in the central nervous system (CNS). Additional progress towards a new therapeutic strategy for GBM need to better understand the mechanisms at the basis of cell growth and chemo-resistance of these cancers. Several reports have suggested that GBM are characterized by high expression of 5-lipoxygenase (5-LO), an enzyme responsible for the biosynthesis of inflammatory molecules that are involved in different pathological processes [9,10,11]. In particular, 5-LO is implicated in the growth and survival of tumor cells, such as different brain tumors, suggesting that inhibition of 5-LO activity could represent a valid drug-target for these cancers. To confirm the implication of 5-LO in the pathogenesis of cancer, researchers have used several medications such as 5-LO inhibitors (zileuton, ZYflo, ABT-761), FLAP inhibitors (MK-886) or other related drugs (zafirlukast and montelukast) in inhibiting cell proliferation and inducing cell death in vitro and in vivo [11]. Our group has synthesized and characterized several agents such as quinone-based derivatives with anti-cancer and anti-inflammatory properties [12,13,14,15,16,17,18,19,20,21,22,23]. These studies on the development of quinone-based compounds with antitumor activity have led to investigate the natural compound embelin and its analogues known as inhibitors of 5-LO and able to suppress proliferation of GBM cells [24]. In this light, we showed that quinone derivative RF-Id, a novel 5-LO inhibitor [16,17,18,19,20,21,22,23,24,25], was able to inhibit the inhibitor of apoptosis proteins (IAPs) and to induce cell death in GBM cells. IAPs are often overexpressed in cancer and play a key role in inhibiting apoptosis blocking caspase activation. A recent study has demonstrated that the hydroquinone-based derivative 3-tridecyl-4,5-dimethoxybenzene-1,2-diol (TDD, EA100 Red) is a very potent direct 5-LO inhibitor [26]. The main aim of this work was to provide a new strategy for GBM treatment. In the light of the potent activity of this novel compound as 5-LO inhibitor, we studied the potential molecular mechanisms at the basis of its anti-tumor effects.

## 2. Results

### 2.1. Effects of EA-100C Red on the Cell Proliferation of GBM Cells

In order to study the antitumor action of the novel hydroquinone-based derivative TDD named EA-100C red, we assessed the effects of the compound on the proliferation of GBM cell lines (U87MG and LN229) after 24, 48 and 72 h of treatment. The inhibition of cell proliferation was detected by MTT assay as described in “Materials and Methods” and occurred in a time- and dose-dependent manner. In particular, afterward 72 h the EA-100C red concentration that inhibited 50% of cell growth (IC_50_) was 75.5 µM in U87MG and 21 µM in LN229 (Figure 1). This compound induced a higher cell growth inhibition on LN229 compared to U87MG cells (Table 1).

Human GBM cells U87MG and LN229 were seeded in serum-containing media in 96-well plates at the density of 2 × 10^3^ cells/well. After 24 h incubation at 37 °C, cells were treated with increasing concentrations of EA-100C red (0.8–100 μM) for 72 h. Cell viability was assessed by a MTT assay as described in Material and Methods. IC_50_ values are reported in Table 1.

### 2.2. Effects of EA-100C Red on Apoptosis 

Interestingly, we investigated the effects of EA-100C red on apoptosis by FACS. U87MG and LN229 cells were treated with EA-100C red at a concentration equal to the corresponding IC_50_ for 72 h and subsequently stained with PI and Annexin V-FITC and analyzed by FACS. 

As shown in Figure 2A, in U87MG, EA-100C red induced late apoptosis in 18% of cell population and early apoptosis in 13% of cell population, as compared to control in which only 8 and 7.4% of cell population was recorded in late and early apoptosis, respectively. On the other hand, in LN229 EA-100C red only increased early apoptosis from 11% of control to 22.7% (Figure 2B). No significant effect on necrosis was recorded. On the basis of the pro-apoptotic effects of EA-100C red, we evaluated its effects on the levels of expression of different genes involved in cell death mechanisms by microarray analysis. After 72 h of treatment with EA-100C red at a concentration equal to the corresponding IC_50_, the low density array analysis showed that the compound induced an up-regulation of several pro-apoptotic genes in both cell lines. In particular, EA-100C red up-regulated NFKBIA and Bcl3, but down-regulated CASP8 in U87MG (Figure 2C). On the other hand, it induced an increase of NFKBIA expression and a decrease of NFKB1 levels compared to control, in LN229 (Figure 2D). EA-100C red thus induced apoptosis by modulating NF-kappaB pathway genes.

### 2.3. Effects of EA-100C Red on Mitochondrial Membrane Potential and Autophagy

To understand the molecular mechanisms underlying apoptosis induced by EA-100C red, we investigated the action on mitochondrial membrane potential (MMP) and autophagy by FACS analysis. First, cells were treated with EA-100C Red at concentrations equal to the corresponding IC_50_ for 72 h and labeled with Mitotracker dye, which emits red fluorescence following ROS accumulation in mitochondria. In particular, after 48 h EA-100C red did not induce a significant increase of MFI in LN229 while % of MFI significantly increased in U87MG (263%) compared to control (Figure 3A,B). After 72 h EA-100C red significantly increased % of MFI in U87MG (237%) and weakly in LN229 (120%) (Figure 3A,B, respectively). H_2_O_2_ was used as positive control. The increase in fluorescence indicated an alteration of the mitochondrial membrane potential. 

Moreover, we studied the effects of EA-100C red on autophagy. Cells were treated with EA-100C Red at a concentration equal to the corresponding IC_50_ for 72 h and subsequently were marked with MDC, that labels autophagosomes. FACS analysis of the data demonstrated that treatment with EA-100C red significantly increased autophagy in both cell lines (235% in U87MG and 225% in LN229) (Figure 4A,B, respectively). Chloroquine was used as positive control. EA-100C red induced disruption of the mitochondrial membrane potential (MMP) and autophagy in both cell lines.

### 2.4. Effect of EA-100C Red on ER-Stress

In order to explain the mode of action by which the compound induces apoptosis and autophagy associated to the alteration of MMP, we studied NF-kappaB-related pathways such as endoplasmic reticulum (ER) stress pathway. In particular, we investigated the effects on gene expression of XBP1 and CHOP that are involved in ER stress signaling. LN229 were treated for 72 h and the expression of XBP1 and CHOP genes was assessed by Real Time PCR. We found that treatment with EA-100C red increased the levels of both CHOP and XBP1 genes but at a higher extent CHOP, compared to the control (Figure 5A). These results confirmed an induction of the endoplasmic reticulum stress by EA-100C red. 

To confirm the induction of ER-stress with EA-100C red treatment, we assessed the emission of fluorescence after staining with the specific dye ER Tracker™ Blue/White. These results showed a significant intensification of fluorescence in the ER of the cells (Figure 5B) after EA-100C red treatment for 72 h. Tunicamycin was used as positive control.

### 2.5. Effects of EA-100C Red on Signal Transduction Pathways 

To elucidate the cell death mechanisms induced by EA-100 C red, we studied its effects on the main signal transduction pathways, the cascade of caspases and the NF-kappaB pathway (Figure 6). In details, the cells were treated with EA-100C red for 24, 48, 72 h as reported in Materials and Methods and the lysates were analyzed by western blotting. In particular, we studied the effects of EA-100C red on caspases 3, 7, 8 and 9; already after 24 of treatment, EA-100C red increased procaspase 8 and decreased the pro-caspase 3 and 9 while it significantly increased the cleaved fragments (Figure 6A). In addition, treatment with EA-100C red already induced after 24 h an increase in Beclin1, p-IKK, p-IKB, p-p65, and p-JNK expression and a significant decrease of the 5-lipoxygenase expression (Figure 6A,B); it increased the expression of p-IRE that is involved in ER-stress mechanism (Figure 6C). In addition, we reported the histograms with the mean of the expression of at least three separate experiments for the investigated proteins in the Appendix A. In conclusion, EA-100C red was probably able to induce ER stress-mediated apoptosis through the activation of caspases and NF-kappaB and JNK pathways.

## 3. Discussion

GBM is the furthermost frequent neoplasia occurring in brain, with unfortunate clinical outcome [2]. In spite of the aggressive conventional treatments, and the latest obtained improvements, GBM regularly recurs and median survival is 14–15 weeks. Additional advances towards a new therapeutic strategy for GBM need to better understand the mechanisms at the basis of cell growth and chemoresistance of these cancers. In the last decade, several approaches have been investigated, such as target-therapies against tumor growth factor receptors, epigenetic regulation, angiogenesis, and metastasis, but there are no evidences that demonstrate patient outcome improvement for this disease [3]. Several benzoquinones have shown antitumor activity on several types of cancer; in particular they are able to regulate apoptosis, cell cycle, production of ROS. Unfortunately, their use is limited by their cardiotoxicity; therefore, our challenge is to develop quinone compounds effective in treating tumors with fewer side effects. A recent work showed that natural benzoquinone derivatives, such as primin and irisoquin show a significant anti-proliferative activity [27,28,29,30,31]. Other important benzoquinones like embelin and its derivatives have shown anti-inflammatory, antioxidant and anti-tumor effects [32,33,34,35,36]. A previous study has demonstrated that the orthoquinone EA-100C and its reduced form EA-100C red, synthesized with a C13 *n*-alkyl chain lacking hydroxyl groups, showed IC_50_ values of 10 and 60 nM, respectively, in cell-free assays representing the most potent 5-LO inhibitors [26].

Our study provides the proof of a complex cross-talk between apoptosis, autophagy, ER stress, and NF-kappaB pathway in GBM cells, which regulates the destiny of cancer cells and senses changes in tumor microenvironment. In response to different stresses, the ER triggers the so called unfolded protein response (UPR), an adaptive response that restores ER homeostasis. If the stress is too long, ER stress activates cell death mechanisms [37]. Further progress towards a therapy for malignant gliomas requires an improved understanding of the mechanisms underlying the switch between pro-survival and pro-death UPR signals and the crosstalk of the UPR itself with other signaling pathways. In this study, we studied the effects of the hydroquinone derivative EA-100C red on growth inhibition of GBM cells. In particular, EA-100C red showed IC_50_ values of 75,5 and 21 μM in U87MG and LN229, respectively, therefore, EA-100C red was more potent on LN229. GBM cells have cancer-specific genetic alterations as described in [38] that could explain this difference in response to the drug.

U87MG, and less LN229, are fragile cells and during detachment (with trypsin or any other method) phosphatidylserine residues in the membrane can translocate and give false positive results; therefore, we always found a 75–80% of live cells even if we used accutase, reduced trypsyn time or scraped cells. According to this, as one can see in the MTT assay that it does not need cell detachment, cell viability is very high. On these bases, we can confirm that low viability does not affect the response to the drug because when cells are gently treated, after detachment we found few false positives. However, considering false positives, apoptosis was significantly increased (two times) with the compound was compared to untreated cells. EA-100C red induced a significant increase in apoptosis and autophagy associated to the disruption of MMP on both cell lines. Probably, the difference of the effects of EA-100C red on the mitochondrial potential is due to genetic differences between the two cell lines and to the different molecular context. In particular, the LN229 cell line seems less sensitive to the effects on ROS production due to its higher antioxidant capacity and expression of enzymes that have a key role in redox homeostasis [39] compared to U87MG. In order to better understand the molecular mechanisms driving its anti-proliferative effects, we assessed the effects of EA-100C red on the expression of genes involved in cell death mechanisms. In particular, EA-100C red increased the levels of NFKBIA expression and decreased NFKB1 expression compared to control, in LN229. On the other hand, in U87MG, EA-100C red up-regulated NFKBIA, Bcl3, but down-regulated CASP8. The difference of caspase 8 expression among the two cell lines is probably not related to a difference in the induction of apoptosis via intrinsic and/or extrinsic pathway; in ER stress both caspases 8 and 9 are often activated. We probably found the difference of caspase 8 expression among the two cell lines because of the different p53 status of the cells.

It is reported that caspase 8 can regulate the NF-kappaB pathway independently from its activity as a pro-apoptotic protease [40]. Probably, EA-100C red induced apoptosis by modulating the NF-kappaB pathway. In this light, we studied NF-kappaB-related pathways such as endoplasmic reticulum stress pathway. In particular, we investigated the effects of compound on the expression of CHOP and XBP1, that are involved in ER-stress mechanisms. In details, EA-100C red induced an up-regulation of XBP1 and at a greater extent of CHOP. Up-regulation of CHOP, triggered by ATF4 induces a persistent state of stress that leads the cells to programmed cell death [37]. Apoptosis and autophagy induced by EA-100C red may be related to induction of endoplasmic reticulum stress. Based on the obtained data, we selected temozolomide-resistant cell line LN229 that resulted more sensitive (IC_50_ = 21 μM) to the drug; this was made in order to better study the molecular mechanism at the basis of the cell death induced by this new compound. It is known that high ROS overproduction leads to ER stress in the cells but it seems that also low levels of ROS can induce protein kinases and phosphatases activation, mobilize deposits of Ca2+, regulate transcription factors, resulting in apoptosis [41]. To investigate if apoptosis induced by EA-100 red could be mediated by ER stress even if ROS levels in mithocondria were low, we selected LN229. Our aim is to find a potential candidate for GBM treatment, for which at least 50% of the patients do not respond to temozolomide and this compound seems to be a useful tool, at least based on our results. We evaluated the effects of EA-100C Red on the main pathways of signal transduction, the cascade of caspases and the NF-kappaB pathway. We studied the p65 subunit of NFKB, and it is not surprising that it is absent in the control by considering that this cell line is p53 mutated and p53 indirectly regulates p65 transcription [42]. 

EA-100C red induced apoptosis through the caspase cascade and led to the activation of IRE-1, JNK, NF-kappaB and the increase of Beclin-1 which are involved in the cell death mechanisms related to the ER-stress. Data coming from literature suggest that an increase of the misfolded proteins can activate ER stress and lead to apoptosis paralleled by autophagy in cancer cells. These effects are regulated by IRE1/JNK/Beclin-1 pathway. Based upon these data, we specifically evaluated the latter pathway [37]. In ER stress conditions, the activation of TRAF2-IRE1 would lead to JNK and NF-kappaB activation that induce apoptosis and autophagy through the activation of caspases and Beclin-1 up-regulation [37]. In conclusion, EA-100C red was probably able to induce ER stress-mediated apoptosis and autophagy associated to the disruption of mitochondrial membrane potential by activating IRE-1 pathway and up-regulating CHOP levels.

## 4. Materials and Methods

### 4.1. Materials

Analytical grade reagents were used in this study and purchased from Sigma Aldrich (Milan, Italy). Carlo Erba silica gel 60 (230–400 mesh; Carlo Erba, Milan, Italy) were used for flash chromatography. We purchased plates coated with silica gel 60F 254 nm from Merck (Darmstadt, Germany) to carry out TLC. We recorded the 1H- and 13C-NMR spectra using an AC 300 instrument (Bruker, Billerica, Massachusetts, USA).

### 4.2. Chemistry

EA-100C red was synthesized as reported in our previous works [13,18,27]. 

### 4.3. Cell Culture 

U87MG and LN229 cells were a gift of Dott. Carlo Leonetti (Regina Elena Cancer Institute, Rome, Italy). LN229 and U87MG were cultured in DMEM and RPMI, respectively, (Life Technologies, Carlsbad, CA, USA) increased with 10% FBS (fetal bovine serum), 1% L-glutamine, streptomycin and penicillin (Lonza Group Ltd., Basel, Switzerland). Cells were grown in a 5% CO_2_ -95% air incubator at 37°C.

### 4.4. Cell Viability Assay

U87MG and LN229 were plated in 96-well plates (2 × 10^3^ cells/well) and treated with increasing concentrations of EA-100C red (0.8–100 μM) for 24 h, 48 h and 72 h. Cell proliferation was evaluated by MTT assay as formerly described [25]. At least three separate experiments were performed.

### 4.5. Analysis of Apoptosis by FACS

Apoptosis was identified by using FITC Annexin V Apoptosis Detection Kit I (BD Biosciences Pharmingen, Heidelberg, Germany). GBM cells were plated in 6-multiwell plates at the density of 2 × 10^5^ cells/well and treated after 24h with EA-100C red. After 72 h, cells were incubated as described by the manufacturers. BD Accuri™ C6, (Becton Dickinson, San Jose, CA, USA) was used to perform FACS analysis. We acquired about 20,000 events for each sample, in at last three separate experiments. Annexin V-FITC and PI fluorescence were detected using the FL1 and FL3 channels, respectively.

### 4.6. FACS Analysis of Mitochondrial Potential

LN229 and U87MG cells were seeded in 6-multiwell plates (2 × 10^5^ cells/well) and treated for 72h. Subsequently, cells were collected and stained with Mitotracker Red probe as described in our previous work [25]. After labeling with the probe, cells were analysed by FACScan, BD Accuri™ C6 and red fluorescence was collected through FL2 channel. We acquired about 20,000 events in at last three separate experiments for each sample.

### 4.7. Flow Cytometric Analysis of Autophagy

GBM cells were plated in 6-multiwell plates (2 × 10^5^ cells/well). Cells were treated for 72 h and then incubated with MDC as described in our previous work [25]. After MDC staining (Sigma, Milan, Italy), cells were collected and analyzed by flow cytometry (FACScan, BD Accuri™ C6). For all the samples MDC fluorescence was collected through the FL1 channel and 20,000 events were acquired in at last three separate experiments. The formula (MFI treated/MFI control) was used to calculate the mean fluorescent intensity.

### 4.8. Taqman Human Apoptosis Array-Real-Time-PCR

After treatment, total RNA was extracted according to the mirVana PARIS (Ambion, Life Technologies) manufacturer’s instructions. Then RNA quantity was quantized by NanodropND-1000 Spectrophotometer (Thermo Fisher Scientific, Wilmington, DE, USA). Reverse transcription was performed according to the QuantiTect Reverse Transcription Kit, (Qiagen, Hilden, Germany) instructions. Taqman human apoptosis array including 93 apoptosis-related genes and 3 housekeeping controls (18S, ACTB, GAPDH) was performed by using ViiA7™ Real time PCR system (Applied Biosystems, Darmstadt, Germany). The comparative cross-threshold (Ct) method was performed to calculate the relative expression of the transcripts. Untreated control was used to normalize treated samples. 

### 4.9. Immunofluorescence with ER Tracker 

The ER stress was identified by staining cells with the ER-specific dye ER Tracker™ Blue/White 1 µM in a solution of PBS, for 30 min at 37 °C. Tunicamycin 2 µM for 16 h represents the positive control. After 72 h treatment with EA-100C red at IC_50_ concentration, cell fixation was performed with 4% paraformaldehyde solution, then permeabilization with 0.1% Triton X/PBS solution, finally blocking was performed with a 1% BSA/FBS solution for 1 h at RT. Images were collected under a fluorescence microscope (LSM 510, X63, Zeiss, Oberkochen, Germany) after MOVIOL counterstaining.

### 4.10. Western Blot Analysis

After 72 h of treatment with the concentration inhibiting 50% of cell growth (IC_50_) of EA-100C Red, cells were collected and lysated for 30 min at 4 °C in 1 mL of lysis buffer (1% Triton, 0.1 NaCl, 0.5% sodium deoxycholate, 10 mM Na_2_HPO_4_, pH 7.4, 1mM EDTA, pH 7.5, 10 mM PMSF, 1 mM leupeptin, 25 mM benzamidin, 0.025 units/mL aprotinin). Total proteins were electrotransferred to a nitrocellulose membrane by using Trans blot turbo (BioRad, Hercules, CA, USA). TBST (150 mM NaCl, 10 mM Tris, pH 8.0, 0.05% Tween 20) was used to wash the membranes. Then, membranes were incubated with specific Abs. The rabbit antibodies raised against, caspase-3, caspase-8, p-IKK, IKB, p-p65, p65, Beclin 1, 5-LO and the mouse antibodies raised against capase-9, p-IKB, IKK were acquired from Cell Signaling Technology (Denvers, MA, USA). The mouse antibodies for p-JNK, JNK and p-IRE were purchased from Santa Cruz Biotechnology (Santa Cruz, CA, USA). Enhanced chemiluminescence detection reagents ECL (Thermo Fisher Scientific, Rockford, IL, USA) was used to develop the blots. Blots were analyzed by using Quantity One software (BioRad Chemi Doc). At least three separate experiments were performed.

### 4.11. Statistical Analysis

All data are expressed as mean ± SD. Analysis of variance (ANOVA) together with Neumann-Keul’s multiple comparison test or Kolmogorov-Smirnov test were performed to obtain statistical information.

## 5. Conclusions

GBM is a highly aggressive brain tumor with fatal clinical outcomes. The current specific anti-tumor therapy is based on three fundamental principles: surgery, chemotherapy and radiotherapy. The medical therapy of brain tumors is still limited and is exclusively entrusted to the administration of temozolomide (TMZ), an alkylating agent that is able to overcome in part the BBB and accumulate in the tumor tissue. Despite new and more specific treatment strategies, relapses reappear in all cases. Based on these considerations, we studied the effects of EA-100C red on the inhibition of U87MG and LN229 cell growth. Moreover, we investigated the effects of the compound on apoptosis and autophagy to identify the molecular mechanism at the basis of the antitumor effects of EA-100C red. In conclusion, EA-100C red induced ER stress- mediated apoptosis together with autophagy. The ER stress increased the levels of CHOP, activated NF-kappaB and JNK pathways that led to the induction of the caspases and Beclin-1 upregulation.

## Figures and Tables

**Figure 1 ijms-20-03836-f001:**
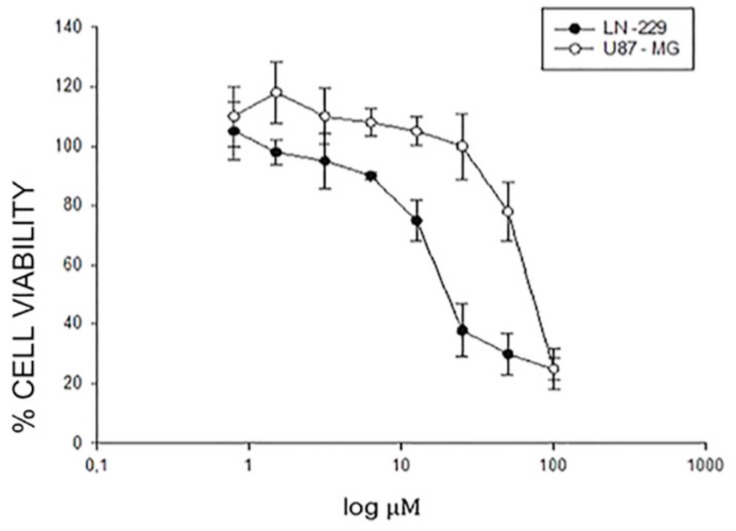
Effects of EA100c red on cell growth inhibition.

**Figure 2 ijms-20-03836-f002:**
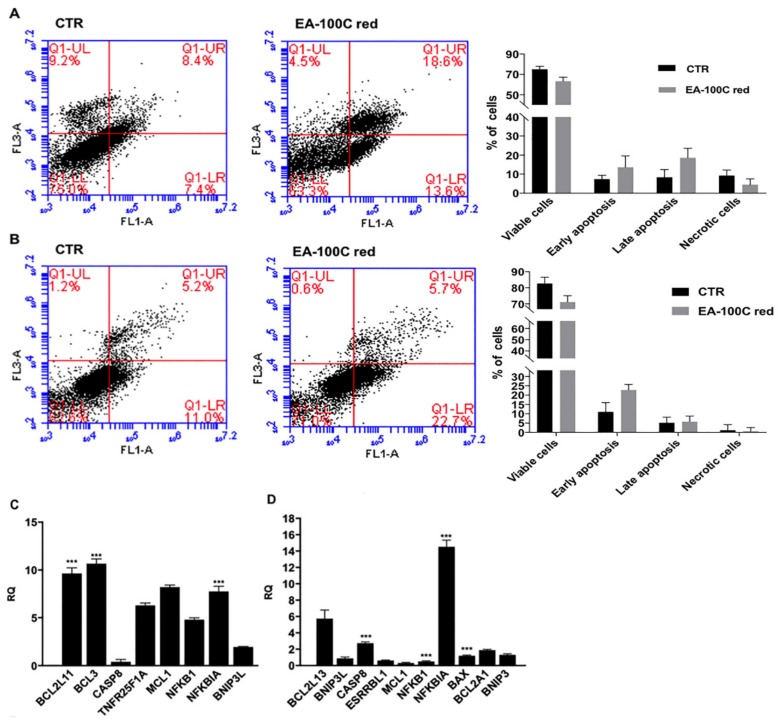
Effects of EA-100C red on apoptosis. (Left) U87MG (**A**) and LN229 (**B**) were not treated or treated with EA-100C red for 72 h at concentrations equal to IC_50_. Apoptosis was evaluated by FACS analysis, after double labeling with propidium iodide (PI) and FITC-Annexin V. The lower left quadrants of each panel show the viable cells, which exclude PI and are negative for FITC-Annexin V binding. The upper left quadrants contain the non-viable, necrotic cells, negative for FITC-Annexin V binding and positive for PI uptake. The lower right quadrants represent cells in early apoptosis that are FITC-Annexin V positive and PI negative. The upper right quadrants represent the cells in late apoptosis, positive for both FITC-Annexin V binding and for PI uptake. The experiments were performed at least three times and the results were always similar. (Right) Analysis of the expression levels of apoptotic genes by qRT-PCR reported as relative quantification (RQ) =2^−ΔΔCt^. Total RNA from U87MG (**C**) and LN229 (**D**) treated with EA-100C red for 72 h. Each experiment was repeated at least three times and data are shown as mean ± SD. *** *p* ≤ 0.001.

**Figure 3 ijms-20-03836-f003:**
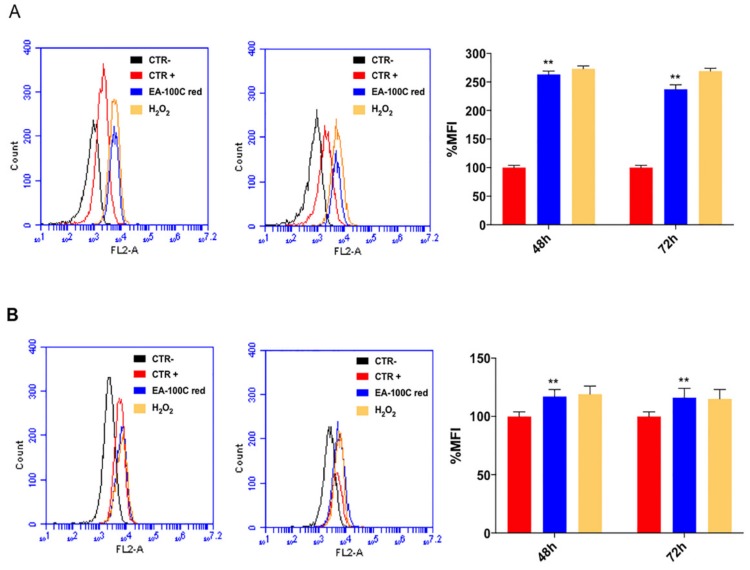
Effects of EA-100C red on mitochondrial membrane potential. Flow cytometric analysis of mitochondrial membrane potential by using Mitotracker Red, a dye which localizes in mitochondria and changes according to mitochondrial membrane potential variations due to oxidative stress. U87MG (**A**) and LN229 (**B**) cells were treated with EA-100C red for 72 h and H_2_O_2_ for 1 h (left panels). CTR+ stained untreated cells, CTR- unstained untreated cells. The % MFIs of control were calculated as described in ‘Materials and Methods’ and represented as columns (right panels). The experiments were performed at least three times and the results were always similar. Bars, SDs. ** *p* ≤ 0.001.

**Figure 4 ijms-20-03836-f004:**
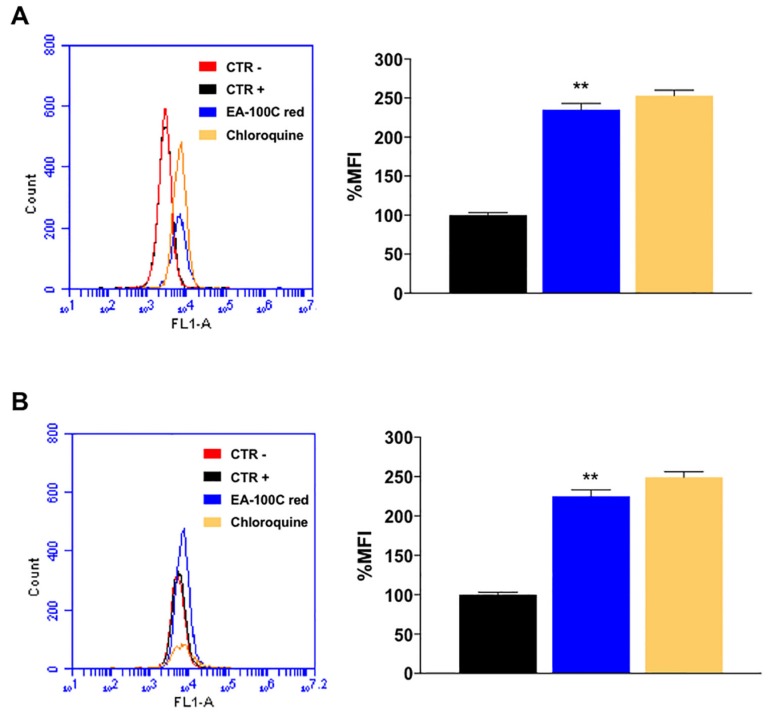
Effects of EA-100C red on autophagy. Flow cytometric analysis of autophagosome formation (MDC incorporation) in U87MG (**A**) and LN229 (**B**) cells treated with EA-100C red and chloroquine for 72 h (left panels). The % MFIs of control were calculated as described in ‘Materials and Methods’ and represented as columns (right panels). CTR+ stained utreated cells, CTR- unstained untreated cells. The experiments were performed at least three times and the results were always similar. Bars, SDs. ** *p* ≤ 0.001.

**Figure 5 ijms-20-03836-f005:**
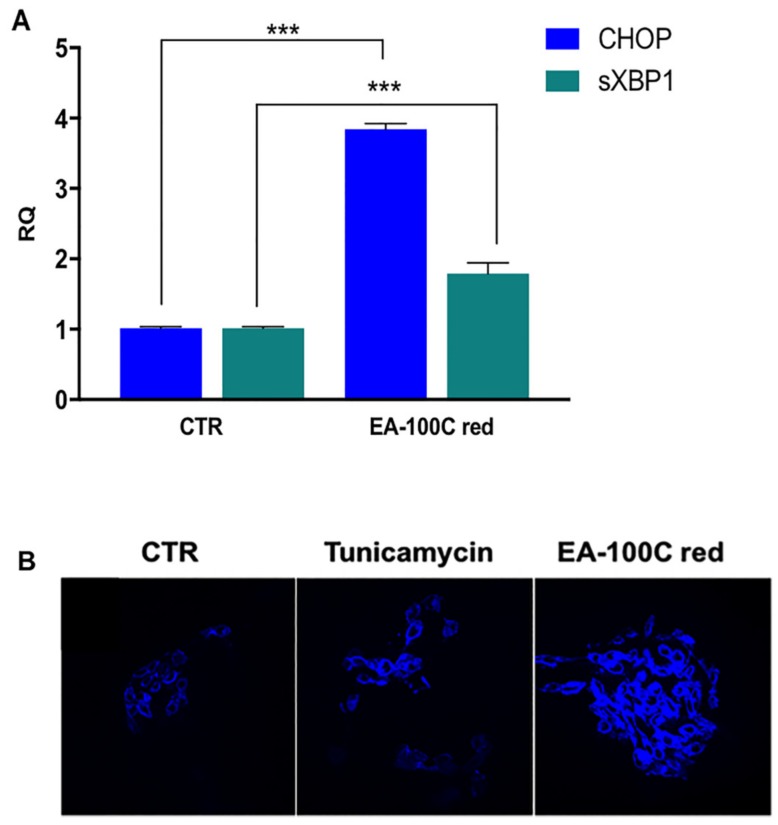
Effect of EA-100C red on the expression levels of XBP1 and CHOP. (**A**) Analysis of the expression levels of XBP1 and CHOP by qRT-PCR using the total RNA from LN229 treated with EA-100C red cells for 72 h. Each experiment was repeated at least three times and data are shown as mean ± SD.*** *p* ≤ 0.01 (**B**) Immunofluorescence with ER-Tracker. LN229 cells were treated with EA-100C red for 72 h. The ER stress was identified by cell staining with the ER-specific dye ER Tracker™ Blue/White 1 µM for 30 min at 37 °C. Tunicamycin 2 µM for 16 h was the positive control.

**Figure 6 ijms-20-03836-f006:**
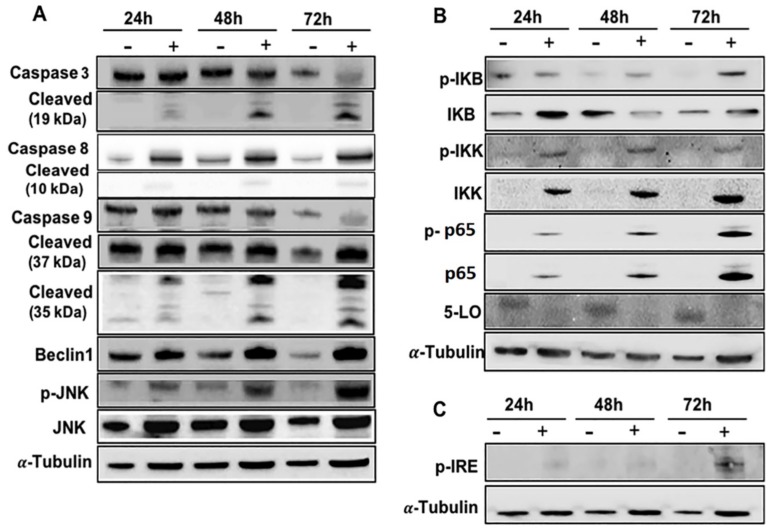
Effects of EA-100C red on signal transduction pathways. LN229 cells were treated for 72 h with IC_50_ of EA-100C red. Cell lysates were incubated with anti-human antibodies and analysed by western blotting; the housekeeping protein α-tubulin was used as loading control. EA-100C red was able to induce already after 24 h the activation of caspases 3 (**A**), 8 (**A**) p65 and 9(**A**), p-JNK(**A**), p-IKK (**B**), p-IKB (**B**), p-p65 (**B**), p-IRE (**C**), the increase of Beclin -1(**A**) expression and the inhibition of 5-LO expression (**B**).

**Table 1 ijms-20-03836-t001:** IC_50_ of EA100C Red on GBM cells.

Name	Structure	U87MG	LN229	*P* Value
**EA-100C Red**	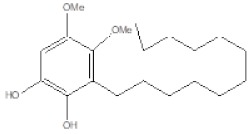	**IC_50_ [µM]** 75.5 ± 0.04	**IC_50_ [µM]** 21 ± 0.05	** *p* ≤ 0.01

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
