# Peer review of "A Hydroquinone-Based Derivative Elicits Apoptosis and Autophagy via Activating a ROS-Dependent Unfolded Protein Response in Human Glioblastoma"

_ijms, 2019, doi:10.3390/ijms20153836_

Round 1

Reviewer 1 Report

The Authors studied the effect of EA-100C red on GBM cell lines. The study could be interesting but the Authors draw very significant conclusions based on a single methods, not always providing conclusive results (eg. induction of apoptosis shown by AnnexinV/PI staining does not match the western blots etc.

  • line 20: anticancer or anti-cancer
  • - Fig. 1 - cell viability rather than cell growth would sound better; MTT assay reflects both proliferation inhibiton and cell death induction.

- Statistical significance in the Fig.1 is confusing. Is it related to difference in IC50 values for both cell lines? If so, I recommend to put it in Table 1 instead of marking at the graph in this way.

- Fig. 2: "The experiments were performed at least three times and the results were always similar" it is not enough! The graph with mean and SD values needs to be included, regarding invisibility of the majority of the percentages on dot plots. In addition, why control cells, especially U87-MG, had low viability? How could it affect the response to drug? AnnexinV/PI staining rather suggests poor capability of the drug to induce apoptosis.

- Fig. 6 - what the Authors mean stating NFKB? Which subunit was immunoblotted? Why NFkB (total) is absent in control cells?

- Fig. 6A - cleaved caspase-8 image seems to be manipulated.

- Fig. 6 - why only one cell line was used for these experiments?

- Was other markers for autophagy tested, eg. LC3B, p62?

- In general, it is not NFKB, it is NF-kappaB. It should be corrected.

- Discussion is very poor.

- the difference between sensitivity of LN-229 and U87-MG cell lines should be discussed.

Author Response

We have very appreciated the comments and suggestions of the referees and we have taken advantages of their considerations. We have tried to answer all their final issues as specified below and we hope now to satisfy their criticisms

Reviewer  1

The Authors studied the effect of EA-100C red on GBM cell lines. The study could be interesting but the Authors draw very significant conclusions based on a single method, not always providing conclusive results (eg. induction of apoptosis shown by AnnexinV/PI staining does not match the western blots etc..

Major comments:

Question: line 20: anticancer or anti-cancer

Answer: According to the reviewer, we have corrected and checked all the text of the revised manuscript.

Question: Fig. 1 - cell viability rather than cell growth would sound better; MTT assay reflects both proliferation inhibiton and cell death induction.

Answer: As suggested, we have corrected y-axis by reporting cell viability.

Question: Statistical significance in the Fig.1 is confusing. Is it related to difference in IC50 values for both cell lines? If so, I recommend to put it in Table 1 instead of marking at the graph in this way.

Answer: As correctly suggested by the reviewer, we have reported p-value in the table because it is related to the difference in IC50 values.

Question: - Fig. 2: "The experiments were performed at least three times and the results were always similar" it is not enough! The graph with mean and SD values needs to be included, regarding invisibility of the majority of the percentages on dot plots. In addition, why control cells, especially U87-MG, had low viability? How could it affect the response to drug? AnnexinV/PI staining rather suggests poor capability of the drug to induce apoptosis.

Answer: As suggested by the reviewer, we have added a histogram including the mean and SD values for each quadrant of the dot plots. U87MG, and less LN229, are fragile cells and during detachment (with trypsin or any other method), phopshatidilserine residues in the membrane can translocate and give false positive results; therefore, we always found a 75-80% of alive cells even if we used accutase, reduced trypsyn time or scraped cells. According to this, as you can see in MTT assay that does not need cell detachment, cell viability is very high. On these bases, we can confirm that low viability does not affect the response to drug because when cells are gently treated, after detachment we found few false positives. However, considering false positives, apoptosis was significantly increased (2 times) with the compound compared to untreated cells.

Question: Fig. 6 - what the Authors mean stating NFKB? Which subunit was immunoblotted? Why NFkB (total) is absent in control cells?

Answer: As correctly suggested by the reviewer, we have specified that it is the p65 subunit of NFKB, and it is not surprising that it is absent in the control by considering that this cell line is p53 mutated and p53 indirectly regulates p65 transcription (Li et al 2015).

Question: Fig. 6A - cleaved caspase-8 image seems to be manipulated.

Answer: Here, (https://www.dropbox.com/sh/c9853ulf69e2vcu/AAAv9QS5hRoCMjhwqwTpCp7Ya?dl=0) we report the original image to demonstrate that it is not manipulated; we used Chemidoc for image acquisition and Chemidoc image can not be modified.

Question: Fig. 6 - why only one cell line was used for these experiments?

Answer: We selected tmz-resistant cell line LN229 that was more sensitive (IC50 = 21 microM) to the drug; this was made in order to better study the molecular mechanism at the basis of the cell death induced by this new compound. It is known that high ROS overproduction leads to ER stress in the cells but it seems that also low levels of ROS can activate protein kinases and phosphatases, mobilize accumulated Ca2+, regulate transcription factors, and result in apoptosis (Chin Chong et al, 2017). To investigate if apoptosis induced by EA100 red could be mediated by ER stress even if ROS levels in mithocondria were low, we selected LN229.  Our aim is to find a potential candidate for GBM treatment, for which at least 50% of the patients do not respond to temozolomide and this compound seems to be a useful tool, at least based on our results.

Question: Was other markers for autophagy tested, eg. LC3B, p62?

Answer: First, we have performed FACS analysis with MDC that labels autophagosomes. Data coming from literature suggest that an increase of the misfolded proteins can lead to the activation of ER stress and induce apoptosis paralleled by autophagy in cancer cells. These effects are regulated by IRE1/JNK/beclin-1 pathway. Based upon these data, we specifically evaluated the latter pathway.

Question: In general, it is not NFKB, it is NF-kappaB. It should be corrected.

Answer: As correctly suggested, we have corrected it.

Question: Discussion is very poor.

Answer: We have improved “Discussion” section of the revised version of the manuscript as correctly suggested by the reviewer.

Question: the difference between sensitivity of LN-229 and U87-MG cell lines should be discussed.

Answer: We have discussed the difference of sensitivity between LN-229 and U87-MG. In fact, the cells have cancer-specific genetic alterations as described in literature (Patil et al, 2015) that could explain the difference in responses to the drug.

We hope that our efforts to address the specific issues raised by the referees have indeed improved the manuscript.

Thank you for your kind consideration and helpful suggestions.

Reviewer 2 Report

The authors describe the anticancer activity of the hydroquinone-based derivative EA-100C red in human glioblastoma cell lines. They unravelled the mechanisms at the basis of its cytotoxic activity based on its ability to induce apoptosis and autophagy. I think the paper deserve publication in the International Journal of Molecular Sciences after the following points will be addressed.

Introduction section:

Line62: quinone-based

Line 63: delete comma between antitumor activity and have

Line 66: please report a brief definition of what IAP proteins are

Line 69: I think it will be useful in the aim to report the name of the analysed compound: EA-100C red

Result section:

Line 95: Mistake in the percentage of early apoptosis of U87MG control that is 7.4% and not 13%

Please harmonize in the text the abbreviations of EA-100C red (i.e. line 103,…)

Figure 2: I think it is necessary to show the histogram reporting the mean of the three experiment performed to assess apoptosis together with the plot

Specify in the legend of Figure RQ abbreviation

Due to the key role of ROS on the pro-apoptotic activity of EA-100C red, I suggest to report the kinetic of intracellular ROS generation after cells treatment, at different time points. 72 h is a long time after treatment and it is interesting to understand what happen after shorter time from hydroquinone-derived exposure.

Figure 3: a positive control is necessary to confirm the good results of the test and should be useful to understand the different effects on mitochondrial potential between the two cell lines. Please report positive control in the histogram. Why CTR+ is perfectly similar to CTR -?

Positive control is necessary also in Figure 4.

The authors performed all the experiments described till this point on the two cell lines. Why only for ER-stress induction and in the evaluation of sgnal transduction pathway they used only LN229. The deeper understanding should be due to the better IC50 of this cell lines, but I think that a comment on the reason why only at this point the authors showed the results on only one cell line is necessary.

Figure 6: Are the results performed only one time? Please specify the number of replicates and if possible showed the histograms with the mean of the expression for the most significant proteins they analysed to support the conclusion of the paper.

Discussion section

Please add references in the discussion section at the end of paragraphs on line 202, 207 and 216.

I think that the difference on the effects of EA-100C red on the mitochondrial potential of the two GBM cell lines deserve an authors’hypothesis in the discussion section.

A discussion on the difference of caspase 8 expression among the two cell lines should follow. The difference could be related to a difference in the induction of apoptosis via intrinsic and/or extrinsic pathway? Please comment. Did the authors performed analysis on the expression of caspase 8 and cleaved caspase 8 in U87MG cells?

Author Response

We have very appreciated the comments and suggestions of the referees and we have taken advantages of their considerations. We have tried to answer all their final issues as specified below and we hope now to satisfy their criticisms

Reviewer 2
The authors describe the anticancer activity of the hydroquinone-based derivative EA-100C red in human glioblastoma cell lines. They unravelled the mechanisms at the basis of its cytotoxic activity based on its ability to induce apoptosis and autophagy. I think the paper deserve publication in the International Journal of Molecular Sciences after the following points will be addressed.

Introduction section:

Question: Line62: quinone-based

Line 63: delete comma between antitumor activity and have

Answer: We have corrected the errors as suggested by the reviewer.

Question: Line 66: please report a brief definition of what IAP proteins are

Answer: According to the reviewer, we have reported a description of IAPs in the revised version of the manuscript.

Question: Line 69: I think it will be useful in the aim to report the name of the analysed compound: EA-100C red

Answer: According to the reviewer, we have reported the name of the compound in the aim.

Result section:

Question: Line 95: Mistake in the percentage of early apoptosis of U87MG control that is 7.4% and not 13%.

Answer: As correctly suggested by the reviewer, we have corrected the mistake.

Question: Please harmonize in the text the abbreviations of EA-100C red (i.e. line 103,…)

Answer: As correctly suggested by the reviewer, we have corrected the abbreviations.

Question: Figure 2: I think it is necessary to show the histogram reporting the mean of the three experiment performed to assess apoptosis together with the plot.

Answer: As correctly suggested by the reviewer, we have included the histogram in the revised version of the manuscript

Question: Specify in the legend of Figure RQ abbreviation

Answer: As correctly suggested by the reviewer, we have specified what RQ means.

Question: Due to the key role of ROS on the pro-apoptotic activity of EA-100C red, I suggest to report the kinetic of intracellular ROS generation after cells treatment, at different time points. 72 h is a long time after treatment and it is interesting to understand what happen after shorter time from hydroquinone-derived exposure.

Answer: We have reported the effects on MMP also after 48h; in U87MG, the compound  induced a significant variation of MMP already after 48h while the effects were lower in LN229.

Question: Figure 3: a positive control is necessary to confirm the good results of the test and should be useful to understand the different effects on mitochondrial potential between the two cell lines. Please report positive control in the histogram. Why CTR+ is perfectly similar to CTR -? Positive control is necessary also in Figure 4.

Answer: We have included positive controls in figure 3 and 4. CTR + in the figure is exactly the same of CTR-, both controls are untreated cells but CTR+ is stained and CTR- is unstained. We have specified this in the figure legend of the revised manuscript.

Question: The authors performed all the experiments described till this point on the two cell lines. Why only for ER-stress induction and in the evaluation of signal transduction pathway they used only LN229. The deeper understanding should be due to the better IC50 of this cell lines, but I think that a comment on the reason why only at this point the authors showed the results on only one cell line is necessary.

Answer: LN229 cell line was more sensitive to the drug but it seemed less sensitive to mithocondrial oxidative stress induction. ROS and ER stress are well known to co-exist and several reports have confirmed that ROS overproduction leads to ER stress in the cells but it seem that also low levels of ROS can activate protein kinases and phosphatases, mobilize accumulated Ca2+, regulate transcription factors, and result in apoptosis (Chin Chong et al, 2017).  In this light, in order to investigate if apoptosis induced by EA100 red could be mediated by ER stress even if ROS levels in mithocondria were low, we selected LN229.  

Question: Figure 6: Are the results performed only one time? Please specify the number of replicates and if possible showed the histograms with the mean of the expression for the most significant proteins they analysed to support the conclusion of the paper.

Answer: As shown in figure, the results were performed after 24, 48 and 72h. According to the reviewer, we report the histograms with the mean of the expression for the investigated proteins.

Discussion section

Question: Please add references in the discussion section at the end of paragraphs on line 202, 207 and 216.

Answer: According to the reviewer, we have added references in the discussion of the revised manuscript

Question: I think that the difference on the effects of EA-100C red on the mitochondrial potential of the two GBM cell lines deserve an authors’ hypothesis in the discussion section.

Answer: We have included our hypothesis in the “Discussion” section of the revised manuscript. Probably, the difference is due to genetic alterations between the two cell lines. In particular, LN229 seems less sensitive to the effects on ROS production due to its higher antioxidant capacity and expression of enzymes that have a key role in redox homeostasis (Zhou et al, 2018) compared to U87MG.

Question: A discussion on the difference of caspase 8 expression among the two cell lines should follow. The difference could be related to a difference in the induction of apoptosis via intrinsic and/or extrinsic pathway? Please comment. Did the authors performed analysis on the expression of caspase 8 and cleaved caspase 8 in U87MG cells?

Answer: The difference is probably not related to a difference in the induction of apoptosis via intrinsic and/or extrinsic pathway; in ER stress both caspases 8 e 9 are often activated.

It is reported that Caspase 8 can regulate the NF-kB pathway independently from its activity as a pro-apoptotic protease (Chaudary 2000); in fact, in the text we stated that the drug modulated NFKB related genes. We probably found the difference of caspase 8 expression among the two cell lines because of the different p53 status of the cells. As reported above we performed western blot analysis only on LN229.

We hope that our efforts to address the specific issues raised by the referees have indeed improved the manuscript.

Thank you for your kind consideration and helpful suggestions.

Round 2

Reviewer 1 Report

Fig. 6A I still have concerns about this figure panel 72 hours cleaved caspase-8 last lane. I did not find such an exposition in the original images attached by the Authors. It is quite clearly visible that the last lane has a very different background.

The other comments have been addressed.

Author Response

Reviewer  1

Question:  Fig. 6A I still have concerns about this figure panel 72 hours cleaved caspase-8 last lane. I did not find such an exposition in the original images attached by the Authors. It is quite clearly visible that the last lane has a very different background.

Answer: We are sorry for this inconvenience. The figure has not been modified as the last lane is more expressed and is darker as you can see in the original images (https://www.dropbox.com/sh/c9853ulf69e2vcu/AAAv9QS5hRoCMjhwqwTpCp7Ya?dl=0). Moreover, all the bands (cleaved  and not) that are overexpressed in the lane are visualized like a smear. Therefore, it seems that they have a different background. Probably, in the preparation of the figure and after cutting, this effect is increased. However, to overcome this problem, we have included a new figure 6 in which we use a caspase 8 with a lower exposure.

We hope now to satisfy your criticisms.

Thank you again for your kind consideration and helpful suggestions .